# Characterization of RNA Sensing Pathways in Hepatoma Cell Lines and Primary Human Hepatocytes

**DOI:** 10.3390/cells10113019

**Published:** 2021-11-04

**Authors:** Wiebke Nicolay, Rebecca Moeller, Sina Kahl, Florian W. R. Vondran, Thomas Pietschmann, Stefan Kunz, Gisa Gerold

**Affiliations:** 1TWINCORE—Centre for Experimental and Clinical Infection Research, Institute for Experimental Virology, 30625 Hannover, Germany; wiebke_nicolay@web.de (W.N.); rebecca.moeller@tiho-hannover.de (R.M.); sina.kahl@icloud.com (S.K.); Thomas.pietschmann@twincore.de (T.P.); 2Center for Emerging Infections and Zoonoses (RIZ), Institute of Biochemistry & Research, University of Veterinary Medicine Hannover, 30625 Hannover, Germany; 3Department of General, Visceral and Transplant Surgery, Hannover Medical School, 30625 Hannover, Germany; Vondran.Florian@mh-hannover.de; 4German Centre for Infection Research (DZIF), 30100 Braunschweig, Germany; 5Institute of Microbiology, Lausanne University Hospital, CH-1011 Lausanne, Switzerland; gisa.gerold@umu.se; 6Department of Clinical Microbiology, Virology, Umeå University, SE-90185 Umeå, Sweden; 7Wallenberg Centre for Molecular Medicine (WCMM), Umeå University, SE-90185 Umeå, Sweden

**Keywords:** hepatoma cells, primary hepatocytes, liver, RNA virus, innate immunity, RIG-I, TLR3, interferon, arenavirus, coronavirus

## Abstract

The liver is targeted by several human pathogenic RNA viruses for viral replication and dissemination; despite this, the extent of innate immune sensing of RNA viruses by human hepatocytes is insufficiently understood to date. In particular, for highly human tropic viruses such as hepatitis C virus, cell culture models are needed to study immune sensing. However, several human hepatoma cell lines have impaired RNA sensing pathways and fail to mimic innate immune responses in the human liver. Here we compare the RNA sensing properties of six human hepatoma cell lines, namely Huh-6, Huh-7, HepG2, HepG2-HFL, Hep3B, and HepaRG, with primary human hepatocytes. We show that primary liver cells sense RNA through retinoic acid-inducible gene I (RIG-I) like receptor (RLR) and Toll-like receptor 3 (TLR3) pathways. Of the tested cell lines, Hep3B cells most closely mimicked the RLR and TLR3 mediated sensing in primary hepatocytes. This was shown by the expression of RLRs and TLR3 as well as the expression and release of bioactive interferon in primary hepatocytes and Hep3B cells. Our work shows that Hep3B cells partially mimic RNA sensing in primary hepatocytes and thus can serve as in vitro model to study innate immunity to RNA viruses in hepatocytes.

## 1. Introduction

RNA viruses from diverse families infect the liver with different outcomes. Some viruses, such as hepatitis C virus (HCV), persistently infect the liver, while others, such as hepatitis A virus, cause acute infections (reviewed in [1]). Beyond such classical hepatitis viruses, many less tissue-specific viruses, including arenaviruses and coronaviruses, replicate in the liver and this contributes to further dissemination and disease development [2,3,4]. As in other organs, the host innate immune system can limit hepatic RNA virus infection by sensing the foreign RNA and establishing an antiviral state in infected and bystander cells [5,6]. This antiviral host response can, in turn, be antagonized by some viruses such as HCV [7,8]. Unless antagonized, innate sensing leads to recruitment of antigen-presenting cells to the site of infection and to activation of adaptive immune responses. Thus, it is critical to understand the innate response of hepatocytes to RNA virus infection.

Viral RNA is sensed by either retinoic acid-inducible gene 1 (RIG-I)-like receptors (RLR) or Toll-like receptors (TLR). While RIG-I-like receptors sense dsRNA in the cytoplasm [9,10], TLR3 and TLR7/8 sense dsRNA and ssRNA [11,12,13], respectively, in endosomal compartments. TLR8 is mainly expressed by monocytes and myeloid-derived dendritic cells [13]. RLR, TLR3, and TLR7, in contrast, are widely expressed in different tissues and cells. Their activation leads to a signaling cascade culminating in the activation of the transcription factors IRF3 and IRF7, which drives the expression of type I and type III interferons (IFN-I, IFN-III). IFN-I, in turn, activates the interferon alpha receptor (IFNAR) and transcription of interferon stimulated genes (ISG) in a paracrine and autocrine manner. ISGs subsequently exert antiviral functions to control infection [14].

Interestingly, all but one of the known hepatitis viruses, i.e., HAV, HBV, HCV, and HDV are highly adapted to the human host, which makes it inherently difficult to study the host response to infection in small animal models [15,16,17]. Mouse models for these viruses either require blunting of the INF-I response to establish an infection as shown for HAV [18], or require tedious xenotransplant models, in which immunocompromised mice are transplanted with human liver cells as used for HBV, its satellite virus HDV, and HCV [17,19,20,21,22]. Notably, these mouse models have the disadvantage that either all murine cells lack the antiviral IFNAR response or that important branches of the immune response are lacking, i.e., B, T, and NK cells. Moreover, transplanted human hepatocytes release human cytokines and chemokines upon virus infection and these are only partially compatible with receptors on murine immune cells. In the HCV field, the partial adaptation of the virus to the murine host [23,24] and the development of surrogate systems for infection, such as rodent hepacivirus infections [25,26,27], hold the promise of fully immunocompetent mouse models for HCV. For the remaining hepatitis viruses, such a goal remains distant to date. Therefore in vitro systems to study the innate immune response to human hepatitis viruses are still needed.

The gold standard of in vitro systems represents primary human hepatocytes (PHH). However, limited availability of human tissue material, donor to donor variability, and rapid dedifferentiation of PHH in vitro hamper the routine use of PHH in the laboratory [28]. Currently, the most promising in vitro system are stem cell-derived hepatocytes [29]. However, the use of embryonic stem cells is ethically banned in several countries and differentiation media—also for induced pluripotent stem cell (iPS) culture and differentiation—is cost-intensive. Thus, PHH and iPS systems are not readily available to many laboratories. Hepatoma cell lines can be considered a cost-saving alternative to study host responses to virus infection.

A major caveat of using human hepatoma cell lines to study innate immunity is that several hepatoma cell lines lack functional RNA sensing pathways. In particular, the HCV susceptible cell clone Huh-7.5 lacks a functional RIG-I pathway [30]. Similarly, TLR3 pathways seem poorly functional in hepatoma cells, having prompted researchers to ectopically express TLR3 in these cells [31]. Alternatively, HepG2 cells were engineered to express CD81 and miR122, two essential host factors for HCV [32]. Moreover, CD81 is an entry factor for the liver stage of the malaria parasite *Plasmodium* spp. Thus, the cell line is suited to study HCV and *Plasmodium* infections and potentially co-infections with both hepatotropic pathogens. How the IFN responses mounted in HepG2-HFL cells upon infection compare to PHH responses is currently unclear. Some studies evaluated the response to synthetic agonists of RNA sensing PRRs in selected hepatoma cells [33,34,35,36]. However, a broad review of RNA sensing capacities of human hepatoma cells lines in comparison to PHH is currently lacking.

In this study, we set out to compare the expression and function of RNA sensors and adaptors in six human hepatoma cell lines versus primary hepatocytes from four donors. Specifically, we tested Huh-6, Huh-7, Hep3B, HepG2, HepG2-HFL, the high fidelity line developed for HCV research [32], and HepaRG cells, which are commonly used in HBV research [34]. As a reference of a cell line derived from a non-tolerogenic organ, we included the airway epithelial cell line A549.

In this study, we observed marked differences in the expression levels of RNA sensors and adaptor molecules in hepatoma cell lines, which was reflected by their RNA sensing capacity. Hep3B cells showed sensing capacities comparable to PHH. Responses to synthetic RNA agonists were confirmed by infection with the RNA viruses human coronavirus 229E (CoV229E) and the arenavirus Tacaribe (TCRV). Taken together, our work suggests that Hep3B cells can, to some extent, mimic PHH innate immunity to RNA virus infection. This is significant as cell lines represent affordable and reproducible in vitro systems, for instance, for antiviral or anti-inflammatory drug screening.

## 2. Materials and Methods

### 2.1. Cell Culture

The hepatoma cell lines Huh-6 [37], Huh-7 [38], HepG2 (ATCC HB-8065), HepG2-HFL [39], Hep3B (ATCC HB-8064), and Vero cells (ATCC CCL-81) were maintained in Dulbecco’s modified Eagle´s medium (DMEM, high glucose, Gibco, Gaithersburg, MD, USA) supplemented with 10% fetal bovine serum (Capricorn Scientific, Ebsdorfergrund, Germany), 2 mM l-glutamine (Gibco), and 1% non-essential amino acids (Gibco). HepG2-HFL cells were cultivated in the presence of 5 μg/mL blasticidin (Gibco) to positively select for CD81 and miR-122 expression. HepG2-HFL cells were kindly provided by Matthew Evans, Icahn School of Medicine at Mount Sinai, USA. For the maintenance of their morphology and polarization, HepG2 and HepG2-HFL cells were cultured in previously collagen R (Serva, Heidelber, Germany)-coated cell culture dishes. The Hepa-RG cell line [40] was cultured in Williams’s E medium (Gibco), supplemented with 10% fetal bovine serum, 2 mM l-glutamine, 5 µg/mL insulin (Life Technologies, Carlsbad, CA, USA), and 50 µM hydrocortisone hemisuccinate (Santa Cruz, Dallas, TX, USA). For the lung epithelial cell line A549 (ATCC CCL-185) F12K medium (Gibco) supplemented with 10% fetal bovine serum and 2 mM l-glutamine was used. The IFN reporter cell line HL116 [41] was grown in DMEM supplemented with 10% fetal bovine serum, 2 mM L-glutamine, and HAT (20 µg/mL hypoxanthine, 0.2 µg/mL aminopterin, 20 µg/mL thymidine, Gibco). HL116 was a kind gift from Sandra Pellegrini (Institut Pasteur, Paris, France). For the generation of Hep3B miR-122 subgenomic replicon (SGR) cells, Hep3B cells were first lentiviral transduced with a lentiviral vector encoding for miR-122 and positively selected with puromycin (2 μg/mL, Sigma-Aldrich, St. Louis, MO, USA) as described earlier [42]. Then, subgenomic RNA, harboring a luciferase reporter gene (Luc-NS3-3′/JFH1), was in vitro transcribed and transfected into the Hep3B miR-122 via electroporation and cultivated in the presence of Geneticin (750 ug/mL, Gibco). All cell lines were incubated at 37 °C and 5% CO_2_.

PHH were obtained from the Department of General, Visceral, and Transplant Surgery at Hannover Medical School. PHH from different donors were isolated from explanted livers and plated directly on collagen-coated plates as previously described [43] and maintained in HCM media (Lonza, Basel, Switzerland). PHH were used immediately after isolation for the respective experiments. Clinical characteristics of the donors are summarized in Appendix A.

### 2.2. PRR Agonist Treatment

To assess the efficiency of TLR3, TLR7/8, TLR4, RIG-I/MDA5, and IFNAR signaling in the different cell lines, cells were pre-treated with IFN-α (100 IU/mL, Schering Corporation) for 24 h or left untreated, followed by stimulation with PRR agonists. The TLR7/8 agonist R848 (Invivogen, Toulouse, France) and the TLR4 agonist LPS (Invivogen) were used at a final concentration of 1 μg/mL or 0.1 μg/mL, respectively, and added directly into the medium. The TLR3 agonist PolyI:C (Invivogen) was added directly to the medium at a concentration of 1 μg/mL or complexed with Lipofectamine 2000 (Thermo Fisher Scientific, Waltham, MA, USA) for transfection for the stimulation of the intracellular PRRs RIG-I and MDA5. Cells were incubated for 6 h or 24 h at 37 °C and assayed for mRNA induction, protein expression, or the supernatants were assayed for IFN-I release.

### 2.3. qRT-PCR

Total RNA was isolated from cell lines using the NucleoSpinRNA kit (Macherey Nagel, Düren, Germany) according to the manufacturer’s recommendations. Total RNA concentration was quantified using a NanoDrop spectrophotometer and stored at −80 °C until further use. Next, 500 ng of the extracted RNA was reverse transcribed using PrimeScript RT Master Mix (Takara, Saint-Germain-en-Laye, France) according to the manufacturer’s recommendations. The transcribed cDNA was then used to determine the upregulation of PRRs in the respective cell lines by qPCR using SYBR Premix Ex Taq II (Takara). Primers for the respective genes were selected from the Harvard Primer Bank (Appendix A). For the reaction, 25 ng of cDNA was used as a template and added to 1 × SYBR Premix Ex Taq II together with each respective primer pair at a final concentration of 0.6μM. The qRT-PCR cycling protocol was set to 45 cycles (10 s at 94 °C, 10 s at 60 °C, and 10 s at 72 °C), followed by a melting curve analysis. Transcript levels were calculated by relative quantification using the ΔΔCt method with GapDH as an internal reference and plotted as 2^−ΔCt^. The qRT-PCR was carried out with the Light Cycler R480 (Roche, Basel, Switzerland).

### 2.4. Immunoblot Analysis

Trypsinized cells were washed with 1 × PBS (Gibco), pelleted, and resuspended in 200 µL of lysis buffer (1% Nonidet P40, 10% glycerol, 1 mM CaCl2 in Hepes/NaCl supplemented with protease inhibitor mix (Sigma #P8340, dilution 1:100). Nuclear debris was removed by centrifugation of the lysate at 12,000× *g* for 10 min at 4 °C. The total protein amount was determined by Bradford assay using RotiQuant Bradford Dye 5 × (Roth, Karlsruhe, Germany). From each sample 50 µg of protein were resuspended in 5 × SDS sample buffer (1.5 M Tris (pH 6.8), 10% SDS, 8% glycerol, 1% β-mercaptoethanol, bromophenol blue). Samples were loaded onto a 12% polyacrylamide-SDS maxi/gel and electrophoresis was carried out overnight at 40 V. Proteins were transferred to a polyinylidenfluoride membrane using the semi-dry Western blot technique. After the transfer, the membrane was blocked in blocking buffer (PBS, 0.5% Tween20 and 5% milk) for 1 h at RT. After blocking, the membrane was incubated overnight 4 °C with the respective antibodies to detect the following proteins: MAVS (mouse α-MAVS clone sc166583, santa cruz 1:100), MDA5 (rabbit α-MDA5 clone D74E4, cell signaling, Danvers, MA, USA, 1:500), RIG-I (rabbit α-RIG-I clone D14G6, cell signaling 1:1000) and TRIF (rabbit α-TRIF clone ab13810, Abcam, Cambridge, UK, 1:500) diluted in 1 × PBS with 0.5% Tween-20 and 1% milk. After the incubation, unbound antibodies were washed off and the membrane was incubated with an HRP-conjugated secondary antibody (α-mouse IgG HRP and α-rabbit IgG HRP, Sigma Aldrich 1:20,000) for 1 h at RT. In addition, the membranes were stained against actin (mouse α-actin/HRP conjugate, santa cruz 1:50,000), as a loading control. The membrane was again washed and the proteins of interest were detected using the ECL Prime Western blotting detection system (GE Healthcare, Chalfont St Giles, UK) according to the manufacturer’s instruction. The proteins were visualized using the ChemoStar Professional Imager System (Intas, Göttingen, Germany). Signal intensity was determined using ImageJ software.

### 2.5. Flow Cytometry Analysis

To measure the surface and intracellular expression of TLR3, TLR7, and TLR8, cells were analyzed by flow cytometry. Therefore, cell suspensions were fixated using 0.5% of PFA in PBS with 1% FCS for 10 min at RT. Afterward, cells were permeabilized with 0.1% Saponin in PBS with 1% FCS for 20 min on ice. Subsequently, cells were stained on ice for 30 min using either primary antibodies against TLR3 (mouse α-TLR3, clone 40C1285.6, novusbio, Wiesbaden Nordenstadt, Germany, 2 μg/1 × 10^6^ cells), TLR7 (rabbit α-TLR7, clone ALX-210-874 enzo, Lörrach, Germany, 2 μg/1 × 10^6^ cells) or TLR8 (mouse α-TLR8, clone 44C143, enzo 2 μg/1 × 10^6^ cells), or an isotype control. Cells were washed with PBS with 1% FCS and stained subsequently with secondary antibodies (α-rabbit-alexa 488, α-mouse-alexa 647) diluted in permeabilization buffer for 30 min on ice in the dark. Cells were washed two times to remove unbound antibodies and finally resuspended in PBS with 1% FCS. Samples were then measured with the C6 Flow Cytometer, BD Accuri (BD Bioscience, Franklin Lakes, NJ, USA) and the analysis was carried out using FlowJo 10 software (Tree Star, Ashland, OR, USA).

### 2.6. Coronavirus Infection

Cells were infected with human CoV-229E-RLuc (kind gift of Volker Thiel) at a MOI 0.1 and cells and supernatants were harvested 24 h later. The cells were washed with 1 × PBS and lysed with 35 µL PBS with 0.5% Triton-X and frozen at −80 °C to ensure complete lysis. Renilla luciferase activity was measured by adding 20 µL of the lysates with 60 µL of luciferase substrate solution (Coelenterazine, Stock: 0.42 mg/mL in methanol, working solution 1:1000 dilution in H20) in 96-well white plates (Berthold). Each well was measured for 0.1 sec in a Microplate reader LB960 CentroX3 (Berthold technologies, Bad Wildbad, Germany) using MicroWin 2000 Software (Mikrotek Laborsysteme, Overath, Germany). The supernatants were UV inactivated and used for IFN quantification as described below.

### 2.7. Arenavirus Infection

Cells were infected with either Tacaribe virus (TCRV) or the vaccine strain of Junin virus (JUNV) Candid#1 at a MOI of 0.01. Supernatant and cell lysates were harvested daily. Supernatants were titrated on Vero cells, followed by immunostaining against the nucleoprotein (MA03-BE06, BEI Resources 1:1500; secondary antibody α-mouse-alexa 488, Invitrogen, 1:500) to determine focus forming units (FFUs). Cell lysates were used for RNA extraction (see above).

### 2.8. Quantification of Bioactive IFN

To quantify the release of bioreactive IFN into the cell culture supernatant, we used the luciferase reporter cell line HL116. This cell line expresses a firefly luciferase gene under the control of the IFN-inducible 6–16 promoter [41]. HL116 cells were incubated with cell culture supernatants from previously stimulated cells for 8 h. Medium was removed, cells were washed with PBS and lysed with 40 µL of Culture Lysis Reagent (Promega). We transferred 10 μL of each lysate into a 96-well white plate. 40 μL/well of substrate (Luciferase Assay System, Promega, Madison, WI, USA ) was added, and each well was measured for 0.1 sec using the Microplate reader LB960 CentroX3 (Berthold technologies) and the MicroWin 2000 Software (Mikrotek Laborsysteme). The amount of IFN-I was calculated based on a recombinant IFN-I standard curve.

### 2.9. Statistics

Statistical analyses were performed using GraphPad Prism software version 8 (San Diego, CA, USA). Specifically, one or two-way analysis of variance (ANOVA) testing was followed by Dunnett, Sidak’s, or Turkey’s multiple comparison test as stated in the Figure legends.

## 3. Results

### 3.1. Hep3B and HepG2 Cells Express Similar Levels of RNA Sensors as Primary Human Hepatocytes

Based on the observation that some human hepatoma cell lines can have dysfunctional RNA sensing pathways [36], we compared six hepatoma cell lines, the lung epithelial cell line A549 and four different PHH donors with regard to the basal expression level of RNA sensors. Transcript levels of the cytosolic dsRNA sensor retinoic acid-inducible gene I (*RIG-I*) showed inter-individual differences of up to two orders of magnitude in the four analyzed primary human hepatocyte cultures (Figure 1a).

*RIG-I* transcript levels in HepG2, HepG2-HFL, Hep3B, and HepaRG were in the range of the primary human hepatocyte transcript levels. In contrast, A549, Huh-6, and Huh-7 cells showed one order of magnitude lower *RIG-I* mRNA levels than PHH. A similar trend was observed for the dsRNA sensor melanoma differentiation-associated protein 5 (*MDA5*) and the RIG-I and MDA5 adaptor mitochondrial antiviral signaling molecule (*MAVS*) with HepG2, Hep3B, and HepaRG expressing similar transcript levels as primary hepatocytes (Figure 1a). The HepG2 subclone HepG2-HFL, which expresses the hepatitis C virus (HCV) host factors microRNA-122 and CD81, showed slightly reduced *MDA5* levels as compared to its parental cell line. Protein expression levels of MDA5 and MAVS largely reflected the respective transcript levels with the exception of low protein levels of MDA5 and MAVS in HepaRG cells (Figure 1b). As basal protein levels of MDA5 and RIG-I were close to the detection limit, we stimulated all cell lines with type I interferon (IFN-I) alpha 2b at 100 IU/mL for 24 h. As expected MDA5 and RIG-I protein levels were markedly increased in all cell lines. Only Huh-7 cells showed low RIG-I protein expression even after IFN-I induction (Figure 1b). Taken together, HepG2 and Hep3B cells expressed cytosolic RNA sensors at levels comparable to PHH.

Next, we addressed expression levels of the endosomal RNA sensors TLR3, TLR7, and TLR8 as well as the TLR3 adaptor TIR domain-containing adaptor protein 1 (TICAM1 or TRIF) in hepatoma cell lines and primary hepatocytes. *TLR3*, *TLR7*, *TLR8*, and *TRIF* transcripts were low in most cell lines (Figure 2a). *TLR7* and *TLR8* levels were markedly higher in PHH as compared to the tested cell lines. TLR3, TLR7, and TLR8 protein were undetectable in most hepatoma cell lines and A549 epithelial cells as measured by surface antibody staining. Only Hep3B and HepaRG cells displayed low levels of TLR3 and TLR8 (Figure 2b–g). TRIF expression on protein level was comparably low in all cell lines (Figure 2h). No inter-individual differences in *TRIF*, *TLR3*, *TLR7*, and *TLR8* transcript levels were observed in PHH from four independent donors.

Finally, we compared transcript levels of interferon-alpha receptor (*IFNAR*) and showed that primary hepatocyte cultures from the four donors had similar expression levels, which were one order of magnitude higher than *IFNAR* levels in the tested cell lines (Figure 3).

### 3.2. Liver Cells Primarily Sense RNA by Cytosolic RIG-like Receptors

Expression of pattern recognition receptors does not necessarily correlate with their function as shown by the dysfunctional RIG-I pathway in the Huh-7 subclone Huh-7.5 [30]. Hence, we stimulated the RNA sensing pattern recognition receptors using synthetic ligands. To mimic uninfected and infected tissue, we left cells untreated or pre-stimulated cells with IFN-I for 24 h, respectively. Next, we stimulated the cells with 1 µg/mL extracellular poly I:C, an agonist of TLR3, and measured mRNA expression levels of *IFN-β* and IFN stimulated genes (*ISG*). Neither the PHH nor the tested epithelial and hepatoma cell lines upregulated *IFN-β* mRNA expression in a TLR3 mediated manner. Only when primed with IFN-I three out of four PHH donors show up to two log increases in *INF-β* transcript levels upon TLR3 stimulation (Figure 4a). In contrast, IFN pretreatment of the cell lines did not sensitize them to the TLR3 ligand (Figure 4a).

The observed induction of *IFN-β* expression was accompanied by induction of *ISG15*, *MxA*, and *RIG-I* at 24 h post-stimulation with extracellular poly I:C in PHH. However, the tested cell lines showed little to no ISG induction upon TLR3 stimulation (Appendix A). Upon IFN priming and TLR3 stimulation, slight induction of *ISG15* was observed in A549 cells as well as slight *MxA* induction in A549, HepG2, HepG2-HFL, and HepaRG cells (Appendix A). The TLR3 mediated ISG induction in primary cells was even higher when cells had been pre-stimulated with IFN-I (Appendix A). This confirms that primary liver cells have a functional TLR3 pathway, which is not fully recapitulated by hepatoma cell lines [35,36].

To test the activation of endosomal ssRNA sensors TLR7 and TLR8, we used the synthetic agonist R848. We failed to detect *INF-β* transcript induction in cell lines or primary cells upon stimulation with 1 ug/mL R848 for 6 h or 24 h, respectively (Figure 4b). The weak induction of ISG transcripts in primary hepatocytes and cell lines (Appendix A) confirmed the refractoriness to ssRNA and the notion that TLR7/8 pathways are mainly active in professional antigen-presenting cells [13].

Finally, we mimicked RLR activation by transfecting the agonist poly I:C into the cytoplasm. PHH robustly responded to RLR stimulation with an up to 100-fold induction of *IFN-I* and ISG induction, independently of previous IFN pre-stimulation (Figure 4c; Appendix A). As expected, A549 cells showed a similarly robust induction of *IFN-I* early after transfection (Figure 4c) and ISGs at a later time point (Appendix A). Huh-6 and Huh-7 were largely refractory to cytosolic dsRNA, and only upon IFN-I priming was a slight induction of *IFN-β* and ISGs was observed (Figure 4c). This is in line with the low basal expression levels of RIG-I and MDA in these cell lines, which can get induced upon IFN-I priming (Figure 1a,b). The other hepatoma cell lines HepG2, HepG2-HFL, Hep3B, and HepaRG showed an order of magnitude stronger *INF-β* and ISG responses than Huh-7 cells. Importantly, PHH responses were in a similar range as those in HepG2, HepG2-HFL, Hep3B, and HepaRG cells (Figure 4c), indicating that these cell lines are suitable in vitro models to study innate immunity to cytosolic dsRNA. Interestingly, IFN-I priming did not enhance RIG-I/MDA5 responses in primary liver cells or HepG2, HepG2-HFL, Hep3B, and HepaRG cells. In contrast, responses in A549 were boosted upon IFN-I priming. Collectively, HepG2, HepG2-HFL, Hep3B, and HepaRG cells best mimicked primary hepatocyte responses to cytosolic RNA stimulation.

### 3.3. A Subset of Hepatoma Cells and Primary Hepatocytes Release Bioactive IFN upon Cytosolic RNA Sensing

*IFN-β* and ISG transcript induction is an indicator for a successfully mounted innate antiviral response. However, the release of bioactive IFN dictates the actual level of the antiviral response. We, therefore, measured bioactive IFN in supernatants of cell lines and primary cells at 24 h post-stimulation with ssRNA and dsRNA agonists. Specifically, supernatants were transferred to HL116 reporter cells, which express luciferase under the control of an IFN-I and IFN-III inducible promoter [41]. The bioactivity assay results largely reflected the observations on the transcript level. PHH released bioactive IFN-I upon TLR3 and RIG-I/MDA5 agonist stimulation, but not upon TLR7/8 agonist stimulation (Figure 5a–c). The effects were independent of previous IFN priming of the primary cells. In the cell lines, only Hep3B cells released IFN upon TLR3 stimulation and only when primed with IFN (Figure 5a). These responses were in the range of PHH responses. We did not observe a response to the TLR7/8 agonist R484; but all cells except Huh-6 and Huh-7 showed robust RIG-I responses (Figure 5b,c). Hep3B cells showed the highest IFN-I release upon RIG-I/MDA stimulation, even higher than A549 cells and PHH. This was again independent of IFN-I priming. In summary, Hep3B hepatoma cell lines showed bioactive IFN release comparable to primary hepatocytes after dsRNA stimulation, confirming them as suitable in vitro models to study hepatocyte immunity to cytoplasmic dsRNA.

### 3.4. Sensing of RNA Viruses in Hepatoma Cells

To analyze innate RNA sensing in an infection setting, we infected primary hepatocytes and all cell lines with the common cold human coronavirus 229E (CoV229E). Coronaviruses are positive-strand ssRNA viruses replicating in the cytoplasm and known to induce IFN [44]. PHH and hepatoma cell lines showed comparable susceptibility to CoV229E, while A549 cells were least susceptible as measured by the activity of a virus-encoded Renilla luciferase (Figure 6a).

Among the hepatoma cell lines, Huh-7 and Hep3B cells showed the highest susceptibility levels comparable to primary hepatocytes (Figure 6a). IFN-I priming reduced coronavirus infection levels slightly with a maximum reduction of one log in primary hepatocytes. We next measured the release of bioactive IFN from CoV229E infected cells. PHH from all donors released IFN upon CoV229E infection (Figure 6c). Most tested cell lines, in contrast, were unable to sense CoV229E and produce IFN. Only A549 and Hep3B cells released bioactive IFN and IFN priming of these cell lines increased CoV229E sensing capacities two- to threefold (Figure 6b). Taken together despite overall low IFN induction in response to coronavirus infection, Hep3B cells seemed to mimic the CoV229E induced IFN release of primary hepatocytes most reliably.

To further test the ability of Hep3B to serve as a cellular model for RNA virus infection, we used Tacaribe virus (TCRV) a member of New World (NW) *Arenaviridae* family. Besides Hep3B cells, we infected A549 cells as a control, as TCRV induces a strong innate immune response in these cells [45,46]. TACV titer in culture supernatants increased over time in both cell lines (Figure 7a) but was one order of magnitude higher in A549 cells as compared to Hep3B cells. Pretreatment with IFN-α decreased viral titer in A549 cells to similar levels as in Hep3B cells, whereas Hep3B cell pretreatment did not affect viral titers. This finding is in line with the previous experiment (Figure 6b), suggesting that IFN priming does not drastically enhance antiviral immunity in Hep3B cells. *IFN-β* induction became apparent at day three post-infection in A549 and increased up to 600 fold on day four compared to uninfected cells, whereas in Hep3B cells, *IFN-β* expression was only slightly induced (Figure 7b).

Arenaviruses efficiently infect the liver [47], thus, co-infections with chronic hepatotropic viruses such as HCV are possible. To evaluate hepatoma cells as a co-infection model system, we mimicked chronic HCV infection in Hep3B cells expressing miR-122 and a subgenomic HCV replicon and infected these cells with either TCRV or the vaccine strain of Junin virus (JUNV) Candid#1 (Figure 8). TCRV titer in Hep3B miR-122 cells increased up to day five, whereas in the presence of the subgenomic HCV replicon TCRV titers decreased starting on day four. A similar effect was observed for JUNV virus infection of Hep3B miR-122 cells, in which the presence of the HCV subgenomic replicon reduced titers from day three onwards. The underlying mechanism of that phenotype is currently under investigation using the Hep3B cell line as a model to investigate potential virus co-infection in the liver.

## 4. Discussion

The liver is a central organ for replication of viruses from diverse families and liver cell immunity is thus a key determinant of infection outcomes. Here we have shown that cultured hepatoma cell lines have a limited ability to sense polyI:C or the RNA viruses such as CoV229E or TCRV despite PRR expression levels, which are comparable to primary hepatocytes. Among the six tested hepatoma cell lines, Hep3B cells displayed the highest sensitivity to cytoplasmic dsRNA stimulation and became sensitive to extracellular dsRNA after IFN-I priming. HepaRG cells sensed intra- and extracellular dsRNA with similar efficiency as Hep3B cells. This is in line with previous studies showing a sensitivity of HepaRG cells to dsRNA [34]. HepG2 cells mounted IFN-I responses via RIG-I only, whereas Huh-7 were refractory to both RIG-I and TLR3 stimulation as previously reported [35,36]. We further show that Huh-6 cells are similarly refractory to dsRNA stimulation as Huh-7 and that HepG2-HFL cells, similar to the parental HepG2 cells, only respond to intracellular dsRNA. This study did not only test immortalized hepatocyte cell lines as reported by Kato and Revill [35,48], but in addition used PHH from four to five independent donors as reference [28,43]. Despite comparable responses of Hep3B cells and PHH to synthetic dsRNA, we observed differential sensitivity of Hep3B cells and PHH to RNA virus infection. While PHH robustly responded to CoV229E infection and released bioactive IFN-I, Hep3B cells only released small amounts of IFN-I. IFN-I priming markedly increased RIG-I and MDA5 expression in Hep3B cells and concomitantly led to increased sensing of CoV229E infection but not of the arenavirus TCRV, suggesting differential viral sensing mechanisms. Taken together, PHH are still the gold standard for studying innate immune responses upon RNA virus infection and Hep3B cells, in particular when primed with IFN-I, may represent a readily available and easy to culture proxy of RIG-I/MDA5 mediated signaling.

Dysfunctional RIG-/MDA5 and TLR3 pathways have been reported in human hepatoma cells [30,36]. This is in line with our observations and correlates with the here reported low expression levels of RIG-I, MDA5, TLR3, as well as the adaptor proteins MAVS and TRIF in Huh-7 and Huh-6 cells. Previous studies overcame these limitations by overexpressing RIG-I, MDA5, or TLR3 and demonstrated sensing of the hepatitis C virus, which was comparable to PHH [31]. For the study of type III IFN responses to the hepatitis C virus, HepG2-HFL cells were developed as a suitable model [32]. Here, we included the HepG2-HFL cell line and showed that IFN-I and ISG responses to RNA ligands remain largely unaltered in HepG2-HFL cells as compared to the parental HepG2 cells. A more recent attractive in vitro system to study innate immunity to hepatotropic viruses are stem cell-derived hepatocytes [49,50,51,52,53]. However, the generation and maintenance of stem cell cultures are resource-intensive and, thus, the system is not available to many laboratories. Consequently, in the absence of immunocompetent animal models for hepatitis viruses, hepatoma cells still qualify as surrogate models to address RIG-I/MDA5 responses and antagonism during RNA virus infection.

In addition to strictly hepatotropic viruses, many human pathogenic RNA viruses target the liver early during infection and replicate in this tissue. Examples are the emerging viruses Rift Valley fever virus and Lassa virus [3,54], which cause liver pathology and thereby may alter adaptive immunity. The detailed contribution of hepatocyte innate immune responses after infection with these viruses remains largely enigmatic to date. In particular, when aiming to study co-infections of these viruses with strictly human tropic hepatitis viruses, the here described cell culture models may aid define the contribution of hepatocyte host responses during infection. Thus, for such defined research questions, there is a justification to use hepatoma cells, which partially mimic RNA sensing, to study hepatocyte infection and superinfection.

## Figures and Tables

**Figure 1 cells-10-03019-f001:**
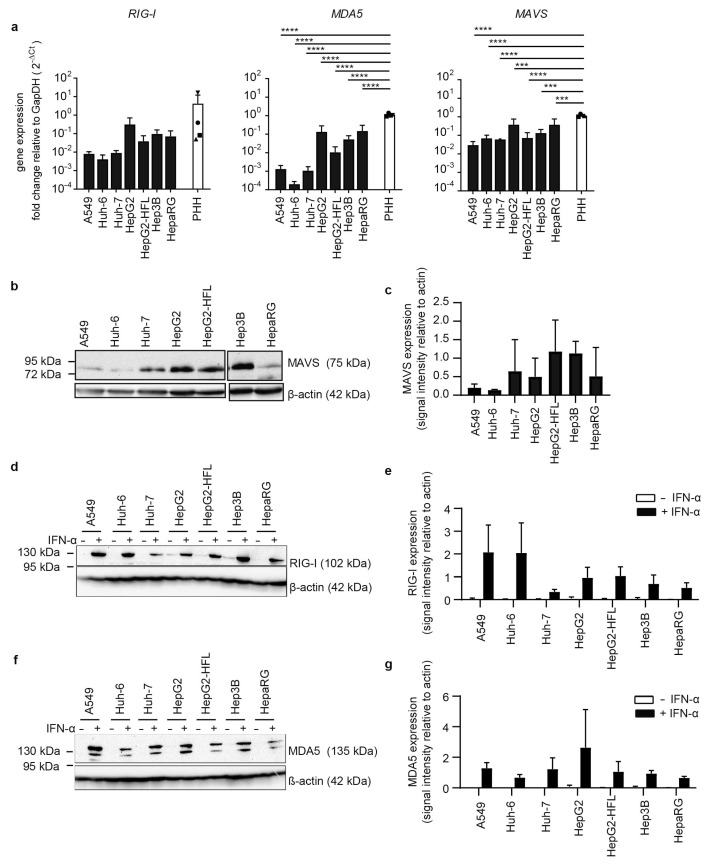
Transcript levels and protein levels of RNA sensors and adaptors in PHH and hepatoma cells. (**a**) RNA expression of *RIG-I*, *MAVS*, and *MDA5* in indicated cell lines and in PHH derived from four different donors. RNA was extracted from whole-cell lysates, mRNA expression measured by qRT-PCR, normalized to *GapDH* mRNA expression, and plotted as 2^−ΔCt^ values. Results are shown as mean ± SD of three independent experiments (with technical duplicates) and in the case of the PHH data were derived from one single experiment (*n* = 1, with technical duplicates). (**b**,**d**,**f**) Immunoblot for MAVS, RIG-I, and MDA5 in lysates from indicated cell lines. For detection of RIG-I and MDA5, cells were left untreated or pretreated with IFN-α (100 IU/mL). β-actin was used as a loading control. Results are representatives of three independent experiments. (**c**,**e**,**g**) Expression levels of the indicated proteins were quantified as immunoblot band density and shown relative to the β-actin loading control as mean ± SD of three independent experiments. One-way ANOVA, followed by Dunnett’s multiple comparison test *** *p* < 0.001, **** *p* < 0.0001.

**Figure 2 cells-10-03019-f002:**
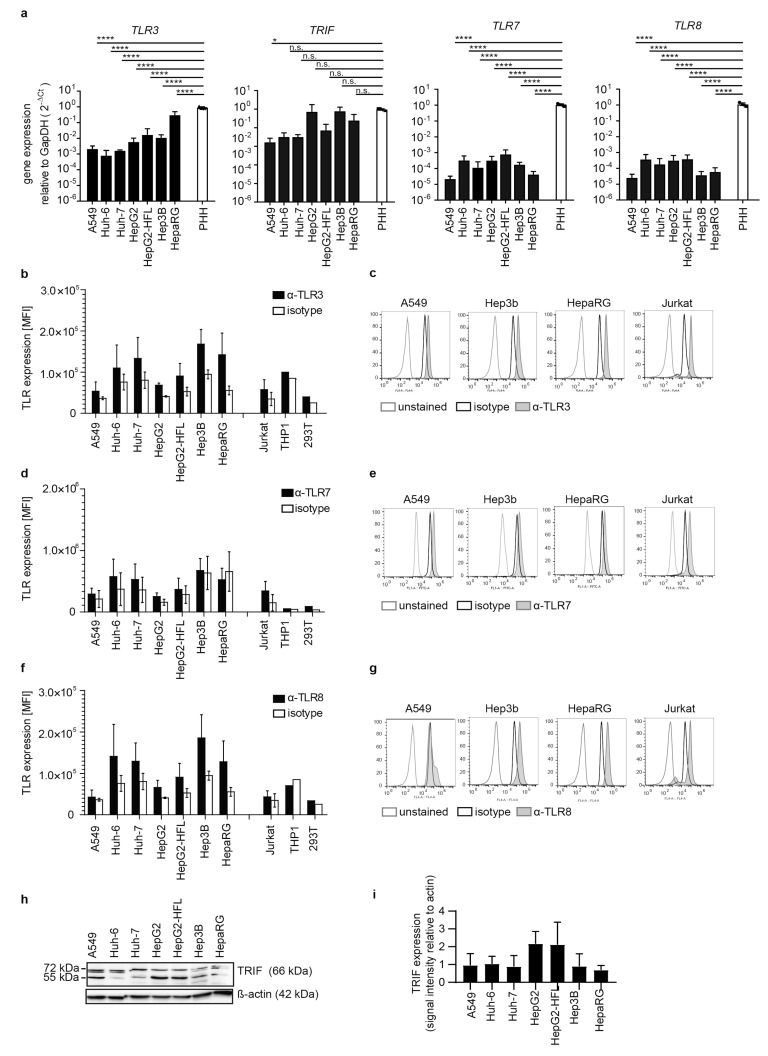
Transcript levels and protein expression of TLRs and the adaptor molecule TRIF in PHH and hepatoma cells. RNA expression of *TLR3*, its adaptor *TRIF*, *TLR7*, and *TLR8* (**a**) in indicated cell lines and in PHH derived from four different donors, as described in Figure 1a. Flow cytometric analysis of intracellular protein expression of TLR3 (**b**,**c**), TLR7 (**d**,**e**), and TLR8 (**f**,**g**) in indicated cell lines. Cells were permeabilized, stained against the respective TLR, an appropriate isotype control, or left untreated. Results are shown as one representative histogram (**c**,**e**,**g**) out of three independent experiments with 30,000 events per measurement for selective cell lines or depicted as MFI (**b**,**d**,**f**) as mean ± SD of three independent experiments. Jurkat cells served as a positive control. (**h**) Immunoblot for TRIF in lysates from indicated cell lines. β-actin was used as a loading control. Results are representatives of three independent experiments. (**i**) Expression levels of TRIF quantified as immunoblot band density and shown relative to the β-actin loading control as mean ± SD of three independent experiments. One-way ANOVA, followed by Dunnett’s multiple comparison test * *p* < 0.05, **** *p* < 0.0001.

**Figure 3 cells-10-03019-f003:**
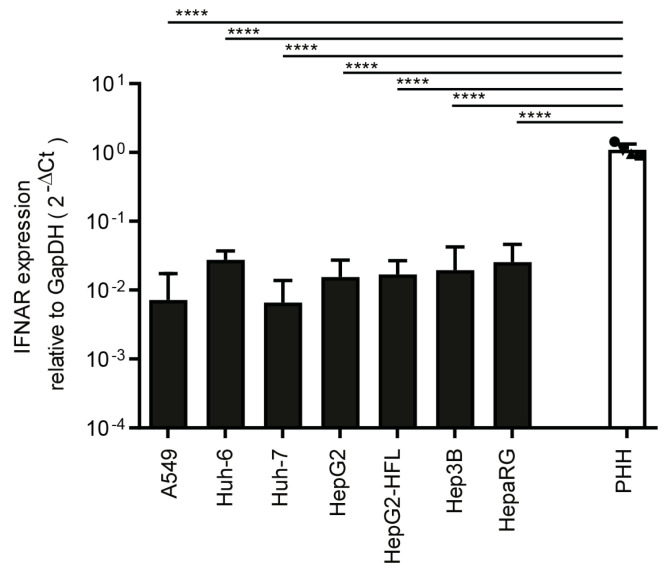
Transcript level of *IFNAR* in PHH and hepatoma cells. RNA expression of IFNAR in tested cell lines and in PHH derived from four different donors. mRNA expression was measured by qRT-PCR as described in Figure 1a. One-way ANOVA, followed by Dunnett’s multiple comparison test **** *p* < 0.0001.

**Figure 4 cells-10-03019-f004:**
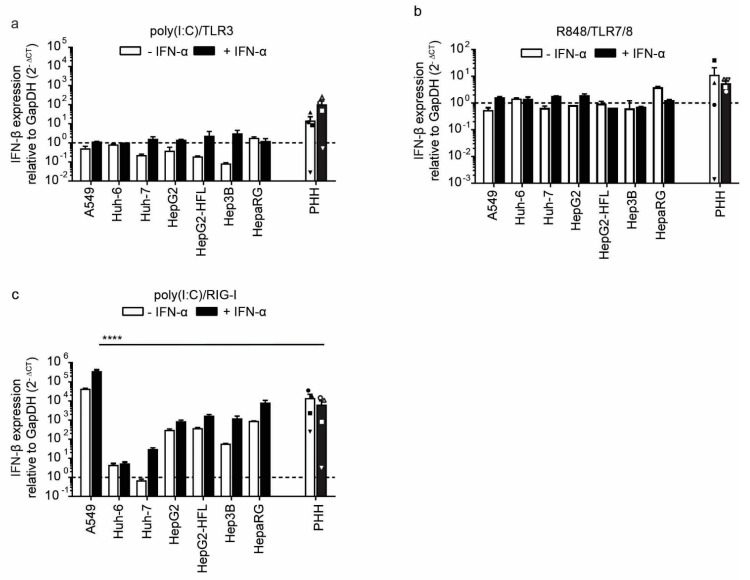
Induction of *IFN-β* expression after stimulation with indicated agonists of RNA sensors in PHH and hepatoma cells with and without *IFN-α* pre-stimulation. *IFN-β* RNA expression in PHHs and hepatoma cell lines mock-treated or pre-treated with IFN-α (100 IU/mL) followed by addition of TLR3 agonist poly(I:C) (1μg/mL) (**a**), TLR7/8 agonist R848 (1 μg/mL) (**b**) or transfected poly(I:C) (2.25 μg/well) (**c**) as RIG-I agonist. Transcript levels were measured by qRT-PCR 6 h after treatment for cell lines and 24 h after treatment for PHH, as described in Figure 1a. Data for the four different PHH donors are shown as mean ± SEM of single experiments performed in technical duplicates for each donor. For the cell lines, mean ± SEM of one experiment (performed in technical duplicates) is shown. Two-way ANOVA, followed by Dunnett’s multiple comparison test, **** *p* < 0.0001. Hepatoma cell lines were tested against PHH and only significant comparisons are indicated.

**Figure 5 cells-10-03019-f005:**
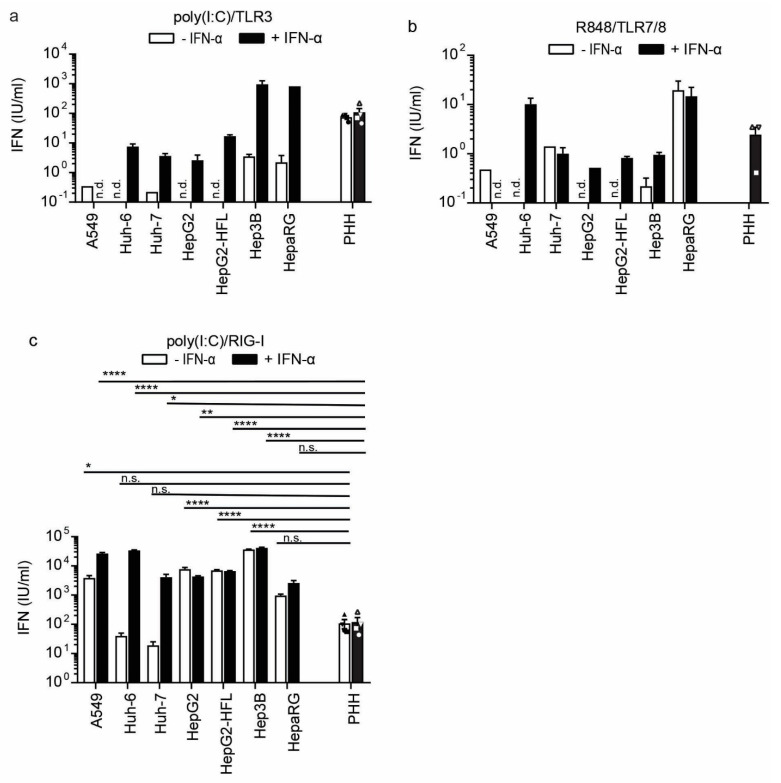
IFN-I secretion from PHH and hepatoma cell lines after stimulation with indicated RNA sensor agonists with and without IFN-α pre-stimulation (100 IU/mL) and further stimulated with TLR3 agonist poly(I:C) (1 μg/mL) (**a**), TLR7/8 agonist R848 (1 μg/mL), (**b**) or transfected poly(I:C) (2.25 μg/well), (**c**) as RIG-I agonist. Released IFN was measured by transferring cell cultures supernatants of stimulated cells to an IFN sensitive luciferase reporter cell line. U/mL was calculated using a recombinant IFN-I standard curve. Data for the four different PHH donors are shown as mean ±SEM of a single experiment performed in technical duplicates and for the cell lines as mean ± SEM of one representative of three independent experiments (performed in technical triplicates). Two-way ANOVA, followed by Dunnett’s multiple comparison test * *p* < 0.05, ** *p* < 0.01, **** *p* < 0.0001. n.d. = not detected.

**Figure 6 cells-10-03019-f006:**
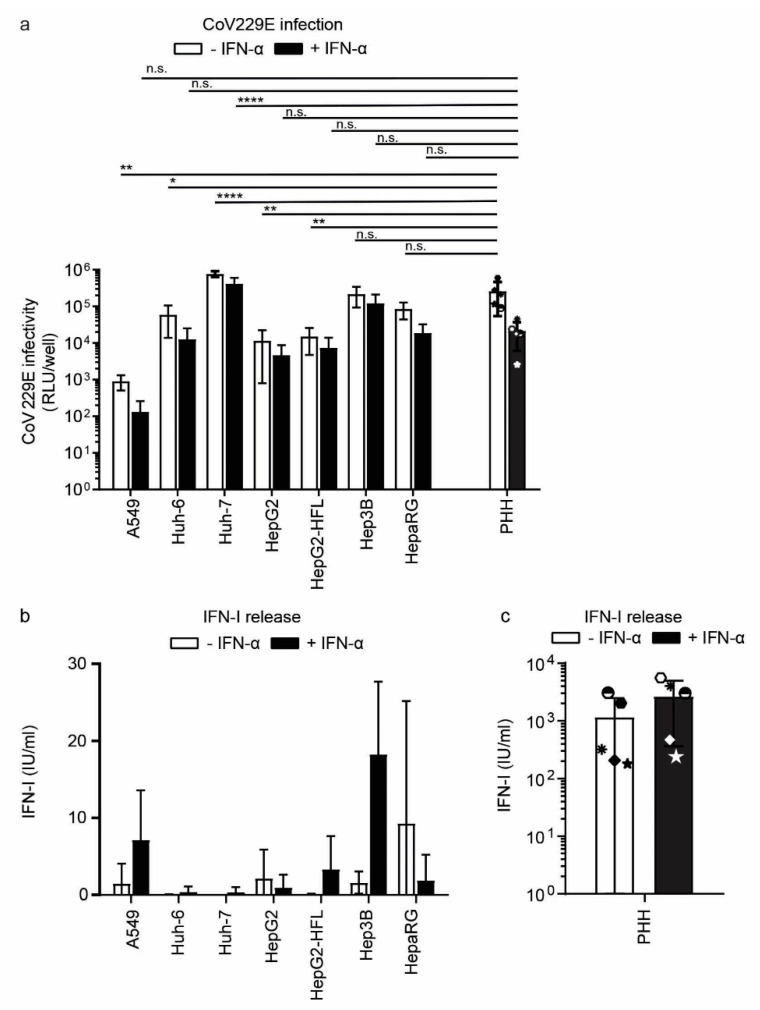
Susceptibility and IFN response of PHH and hepatoma cells to CoV229E. (**a**) CoV infection in PHH and hepatoma cell lines. Cells were mock-treated or pretreated with IFN-α for 24 h and subsequently infected with CoV Renilla reporter virus (MOI 0.1) for 24 h. Infection with the reporter virus was determined 24 h post-infection by measuring luciferase activity. (**b**) IFN-I secretion from hepatoma cell lines and PHH (**c**) mock-treated or pretreated with IFN-α for 24 h followed by CoV infection. Released IFN was measured as described in Figure 5. Data for the five different PHH donors are shown as mean ±SD of a single experiment performed in technical duplicates and for the cell lines as mean ± SD of three independent experiments (performed in technical triplicates). Two-way ANOVA, followed by Dunnett’s multiple comparison test * *p* < 0.05, ** *p* < 0.01, **** *p* < 0.0001.

**Figure 7 cells-10-03019-f007:**
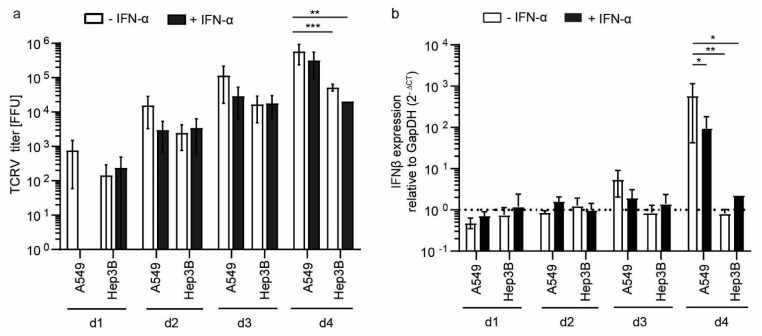
Susceptibility and IFN response of A549 and Hep3B cells to TCRV. Cells were mock-treated or pretreated with IFN-α for 24 h and subsequently infected with TCRV (MOI 0.01) for indicated time points (**a**). Viral titer was determined by titration on Vero cells and staining against the nucleoprotein. *IFN-β* mRNA expression was measured by qRT-PCR as described in Figure 1a (**b**). Results are given as a mean ± SEM of three independent experiments (with technical duplicates). Two-way ANOVA, followed by Turkeys’s multiple comparison test * *p* < 0.05, ** *p* < 0.01, *** *p* < 0.001.

**Figure 8 cells-10-03019-f008:**
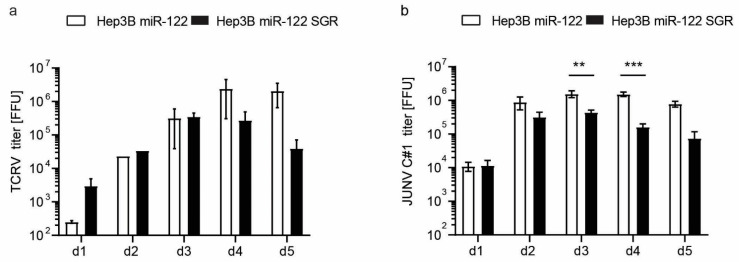
Susceptibility of Hep3B miR-122 and Hep3B miR-122 subgenomic HCV replicon cells to TCRV (**a**) and JUNV Candid#1 (C#1) (**b**). Cells were infected with indicated arenaviruses at an MOI of 0.01 and supernatants were harvested at indicated time points. Viral titers were determined by titration on Vero cells and staining against the nucleoprotein. Results are given as a mean ±SEM of three (JUNV) or two (TCRV) independent experiments (with technical duplicates). Two-way ANOVA, followed by Sidak’s multiple comparison test ** *p* < 0.01, *** *p* < 0.001.

## Data Availability

Data available on request.

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
