# Peer review of "Characterization of RNA Sensing Pathways in Hepatoma Cell Lines and Primary Human Hepatocytes"

_cells, 2021, doi:10.3390/cells10113019_

Round 1

Reviewer 1 Report

Manuscript submitted by Nicolay and co-authors offer analysis and comparison of RNA sensing properties in 5 different hepatoma cell lines in comparison with primary human hepatocytes (and with primary airway epithelial cell line A549 as neg control). The major strength in such comparative analysis stands exactly in the decision to use primary human hepatocyte in comparison with established cell lines. Unfortunately, no data on pos control PHH has been offered. Age and gender for the donor where PHH have been isolated from may be instrumental. As well as the information if PHH have been exposed to cryogenic preservation or analyzed immediately after isolation.

An additional hepatoma cell line, HepG2-HFL subclone, where microRNA-122 and CD81 have been over-expressed, has also been included in the comparative analysis. However, the reason behind this decision may be quite questionable and probably deserves a little bit of discussion.

The manuscript is interesting but sometimes redundant and particularly descriptive. The authors should consider shortening up and revise current version, particularly Results section

The first 7 lines in the Results are actually introducing the study and supporting the aim of the experimental settings. Thus, they should be moved into the Introduction.

In all Figures, the PHH measurements can be combined. If there is any particular reason why such pos control cannot be combined, such statement should be crystal clear from the beginning and specified in the Introduction (and supported by donor characteristics)

Figures 4, 5 and 6, panel a and b, c and d, e and f, can be combined as previous Figures 1-3.

Histogram with dots over-imposed should be offered to evaluate variability. P values need also to be included. When statistical relevance is not reached, interpretation or speculative conclusion should be carefully offered and pondered.

Overall, the manuscript is well written and clear. Minor modification and a final critical revision would easily define a final version ready for publication

Reviewer 2 Report

In the current study, authors focused on the caracterization of RNA sensing pathways in hepatoma cell lines and primary human hepatocytes. It was an interesting research and has a scientific contribution to the related research field. However, there are some issues to be figured out before publication. My comments are as follows:

  1. Please add the significance and novelty of your study in the last part of the Introduction.
  2. Line 109, please add the sources of these hepatoma cell lines.
  3. Line 110, did these cell lines use the same DMEM medium? high glucose or low glucose? Please give the details.
  4. The vendors of some regents or materials were missing. Please add it.
  5. The accession number for each gene need to be added in Table S1. The sequence for each primer was 3'-5' or 5'-3', which should be clarified.
  6. Please improve the method of qRT-PCR. The present vision was too simple and general.
  7. Line 154, there was a typo.
  1. Why did authors not perform the statistical analysis? If not, how did authors evaluate the differences among the experimental groups?
  2. Authors did not give the optical density of all the blots, just showed a representative blot. Reviewer notice that the expression of β-action was not equal among the experimental groups. Please explain.
  3. For PHHs, only 4 donors and no duplication for each donor, the obtained data was not solid.

Author Response

In the current study, authors focused on the characterization of RNA sensing pathways in hepatoma cell lines and primary human hepatocytes. It was an interesting research and has a scientific contribution to the related research field. However, there are some issues to be figured out before publication. My comments are as follows:

  1. Please add the significance and novelty of your study in the last part of the Introduction.

We moved the first sentences from the results sections into the introduction (now lines 101-105).

Moreover we added the significance to the end of the Introduction (lines 109-110):

“This is significant as cell lines represent affordable and reproducible in vitro systems for instance for antiviral or anti-inflammatory drug screening. “

  1. Line 109, please add the sources of these hepatoma cell lines.

We added references to each cell line (lines 121-123). Please see track changes version of the manuscript.

  1. Line 110, did these cell lines use the same DMEM medium? high glucose or low glucose? Please give the details.

The cell lines Huh-6, Huh-7, HepG2, HepG2-HFL, Hep3B and Vero cells were maintained in DMEM high glucose. We added this information in line 123

  1. The vendors of some regents or materials were missing. Please add it.

We added the missing information. Please see track changes version of the manuscript.

  1. The accession number for each gene need to be added in Table S1. The sequence for each primer was 3'-5' or 5'-3', which should be clarified.

We thank the reviewer for pointing this out. We added the UniProt accession number for each gene and the orientation of the primers. Please note that the primers are now listed in Table S2.

  1. Please improve the method of qRT-PCR. The present vision was too simple and general.

We thank the reviewer for this helpful comment. The qRT-PCR method part was amended and primer concentrations and cycling protocol settings were added.

  1. Line 154, there was a typo.

Thank you for pointing this out. We removed the superfluous bracket 

  1. Why did authors not perform the statistical analysis? If not, how did authors evaluate the differences among the experimental groups

The reviewer is absolutely correct and we agree that statistical evaluation of the differences between the cell lines and the PHH is important to convey the message of the study. We now performed statistical analysis and compared the cell lines with the combined data for the PHH donors (see response to reviewer #1 on page 2 of this point-by-point response). We added the statistical analysis to each figure and describe it in the methods section (lines 244-248) and the figure legends.

  1. Authors did not give the optical density of all the blots, just showed a representative blot. Reviewer notice that the expression of β-action was not equal among the experimental groups. Please explain.

Point well taken. We actually repeated the blots and also screened all previous blots. The majority of blots showed equal β-actin expression and we now show a more representative blot.

  1. For PHHs, only 4 donors and no duplication for each donor, the obtained data was not solid.

We used four to five different primary hepatocyte donors, which is quite common in the field (please see (Zhang et al., 2020; Zapatero-Belinchón et al., 2021). We performed each PHH experiment in technical duplicates but as PHH are primary cells and de-differentiate rapidly ex vivo, it is not possible to repeat the experiments with PHH from the same donor in several biological replicates. Instead, it is common practice in the field to compare PHH from multiple donors and the here analyzed number of donors is typically deemed statistically sound.

Additional changes:

  • Due to incorporation of the clinical characteristics of the PHH donors in Table S1, the original Table S1 including the qPCR primer is now labeled Table S2. This was amended in the text accordingly.

References:

Zapatero-Belinchón FJ, Ötjengerdes R, Sheldon J, Schulte B, Carriquí-Madroñal B, Brogden G, et al. Interdependent Impact of Lipoprotein Receptors and Lipid-Lowering Drugs on HCV Infectivity. Cells 2021; 10

Zhang Z, Trippler M, Real CI, Werner M, Luo X, Schefczyk S, et al. Hepatitis B Virus Particles Activate Toll-Like Receptor 2 Signaling Initially Upon Infection of Primary Human Hepatocytes. Hepatology 2020; 72: 829–44.  

Round 2

Reviewer 2 Report

Reviewer highly recommends authors to add the optical density analysis for all the blots.
